

# *Chaetomium globosum* from *Alisma orientale* (Sam.) Juzep. enhances the antioxidative stress capacity of *Caenorhabditis elegans*

Nayu Shen[1,*], Zhao Chen[2,*], Siyu Wang[1], Mingqi Zhang[1], Yujie Jia[1], Xinyu Zhang[1], Yirong Xiao[3], Zizhong Tang[1], Qingfeng Li[1], Ming Yuan[1] and Tongliang Bu[1]

[1] College of Life Sciences, Sichuan Agricultural University, Ya'an, China
[2] Ya'an People's Hospital, Ya'an, China
[3] Sichuan Agricultural University Hospital, Ya'an, China
[*] These authors contributed equally to this work.

## ABSTRACT

**Background**. Medicinal plant endophytic fungi hold significant potential for producing natural antioxidants, as they thrive in environments rich in bioactive antioxidant compounds.

**Methods**. This study focuses on *Chaetomium globosum*, an endophytic fungus isolated from *Alisma orientale* (Sam.) Juzep., to explore the *in vivo* antioxidant activity associated with the ethyl acetate extract (CGE) and to explore the underlying mechanisms.

**Results**. The results indicated that CGE treatment enhances the stress resistance in *Caenorhabditis elegans*, stimulates the antioxidant protection mechanisms of *C. elegans*, and exhibits strong antioxidant activity *in vivo*. RNA-seq analysis showed that CGE regulates Fatty acid degradation, glutathione metabolism, longevity regulating pathway-multiple species and MAPK signaling pathway in *C. elegans*. This study offers an important reference to the utilization of CGE as an antioxidant in the food and medical sectors, while also establishing a theoretical framework for utilizing the *C. globosum* as a natural antioxidant source.

## INTRODUCTION

Oxidative stress arises from an imbalance between pro-oxidant and antioxidant systems in the body, and is strongly associated with the progression of numerous diseases and human aging processes (*Liguori et al., 2018*; *Gulcin, 2020*; *Picardo & Dell'Anna, 2010*; *Marx, 1987*). Given the increasing demand for a healthy lifestyle, research on antioxidants has become a prominent area of investigation both domestically and internationally (*Varesi et al., 2022*).

Consuming exogenous antioxidants can enhance the body's ability to combat oxidative stress, representing a key mechanism for antioxidant defense (*Nwozo et al., 2023*). Synthetic antioxidants raise concerns due to toxic side effects and environmental pollution. Thus,

Corresponding author
Zizhong Tang, 67031988@qq.com

natural antioxidants are increasingly favored for their safety, efficacy, and potential against human diseases and aging (*EFSA Panel on Additives and Products or Substances used in Animal Feed et al., 2019*; *Calokerinos et al., 2023*; *Ramana et al., 2018*; *Xu et al., 2017*). Natural phenolic compounds exhibit potent free radical scavenging capabilities due to their multiple phenolic hydroxyl groups. They represent a significant portion of plant secondary metabolites and serve as the primary source of natural antioxidants (*Nwozo et al., 2023*; *Merecz-Sadowska et al., 2021*; *Halake, Birajdar & Lee, 2016*). The primary method for obtaining natural antioxidants is direct isolation of bioactive compounds from plants, particularly medicinal plants (*Xu et al., 2017*; *Fernandes et al., 2023*; *Zhang et al., 2022b*). Nevertheless, this approach presents challenges such as low extraction efficiency, high costs, and the depletion of plant and land resources. Thus, the development of novel methods for producing natural antioxidants is crucial.

Endophytic fungi represent one source of natural antioxidants (*Caruso et al., 2022*). Endophytic fungi are abundant in various metabolites, including phenols, alkaloids, organic acids, and terpenoids, which possess antibacterial, antioxidant, anti-diabetic, and anti-tumor properties (*Tousif et al., 2023*; *Wen et al., 2022*; *Ortega et al., 2021*). Certain endophytic fungi found in plants possess the capability to produce active compounds that are identical or similar to those of their host plants, and they can synthesize these substances at a faster rate than the host plants themselves (*Jia et al., 2016*; *Hashem et al., 2023*; *Gu et al., 2022*). Moreover, the fermentation of active substances by endophytic fungi offers advantages such as a short production cycle, reduced expenses, elevated output, and conditions that can be easily regulated (*Tang et al., 2022*). The utilization of endophytic fungi fermentation for synthesizing natural antioxidants presents a promising application prospect. The endophytic fungi found in medicinal plants show great potential to serve as a source and pathway for natural antioxidants, which is attributed to their unique living environment abundant in antioxidant components (*Gupta et al., 2020*; *Eshboev et al., 2023*). Currently, a wide array of research findings suggest that isolating of antioxidant-active endophytic fungi from medicinal plants is promising. The endophytic fungus *Penicillium oxalicum* was isolated from the medicinal plant *Ligusticum chuanxiong* Hort by *Tang et al. (2022)*. Their research revealed that the fermentation broth extract of *P. oxalicum* protects lymphocyte DNA from damage and enhances the ability of *C. elegans* to withstand oxidative stress. The bacterium *Pestalotiopsis neglecta*, extracted by *Almustafa & Yehia (2023)* from *Ziziphus spina-christi* M, demonstrated potent neutralizing properties against the free radical 2,2 -diphenyl-1-picrylhydrazyl, offering substantial DNA defense against harm caused by hydroxyl radicals and serving as a potential antioxidant.

In earlier studies conducted by our research group, *Chaetomium globosum*, isolated from the medicinal plant *Alisma orientale* (Sam.) Juzep, exhibited promising potential as a natural source of antioxidants (*Shen et al., 2023*). The extract of *C. globosum* (CGE) exhibits a high total phenol content and demonstrates strong antioxidant activity *in vitro*. However, there are still gaps in the research regarding the antioxidant capacity and mechanisms associated with CGE *in vivo*. Complete sequencing of the *C. elegans* genome has been accomplished, uncovering that approximately 60 to 80% of its genes are homologous to those found in humans. Several advantages make the model organism *C. elegans* ideal for

research, including its non-toxic and benign nature, short lifespan, rapid reproductive cycle, ease of culturing and observation, and the ability to self-fertilize resulting in over 300 offspring. Consequently, *C. elegans* has emerged as a widely used model organism for investigating antioxidative stress and mechanisms related to aging (*Li et al., 2024*; *Zhu et al., 2022*). This research seeks to explore the antioxidant properties and basic principles of CGE *in vivo* by utilizing *C. elegans*. The goal is to establish a theoretical foundation for utilizing *C. globosum* as a natural antioxidant source and to potentially enhance the application of CGE as an antioxidant in the fields of food and medicine.

## MATERIALS AND METHODS

### Materials

After being isolated from the roots of *Alisma orientale* (Sam.) Juzep., the endophytic fungus *Chaetomium globosum* was kept in the Fermentation Engineering Laboratory of the College of Life Sciences at Sichuan Agricultural University, Ya'an, China. The isolation and screening of *C. globosum* were described in detail in a previous publication by *Shen et al. (2023)*. CGE preparation was performed as previously described (*Shen et al., 2023*). Briefly, *C. globosum* was fermented in PDB medium for 7 days, followed by vacuum filtration to remove mycelia and spores. The filtreted fermentation solution was extracted with ethyl acetate, and the extract was concentrated *via* rotary evaporation before lyophilization to obtain CGE (lyophilized powder). The CGE was reconstituted in DMSO to a 200 mg/mL stock; except for acute toxicity assays (diluted in M9 buffer), all experimental concentrations refer to CGE in solid nematode growth medium (NGM).

### Strains of *Caenorhabditis elegans* and their cultivation conditions

The wild-type N2 and *Escherichia coli* OP50 strains were procured from the *Caenorhabditis* Genetics Center located in the United States. In a 20 °C incubator, the worms were grown on NGM plates seeded with *E. coli* OP50. The bleaching method was employed to synchronize the age of the nematodes, during which eggs were obtained by applying a lysis solution (5% NaClO was mixed with 0.5 M NaOH at a 1:9 ratio and prepared fresh) to adult worms. Unless otherwise specified, the synchronized worms were placed on NGM plates containing CGE and *E. coli* OP50 for cultivation at 20 °C. The blank control was CGE free, and the positive control was resveratrol (Res, 20 μg/mL).

### Acute toxicity assay

To minimize the effect of DMSO on worm lifespan and overall health, CGE was diluted in M9 buffer to final concentrations of 10, 40, 70, 100, 150, and 200 μg/mL, ensuring that the DMSO content remained below 1% (*Chen et al., 2022*). Synchronized L4-stage worms were placed in 96-well plates containing 200 μL CGE solution per well, with M9 buffer as a blank control. After incubating at 20 °C for 24 h, the number of surviving worms was counted, and the results were subsequently expressed as a survival rate. The experiment was conducted thrice, with 60 worms randomly selected for each group. Survival rate % = (number of alive worms)/total number of worms ×100%.

## Stress resistance assessment
### Juglone-induced oxidative stress assay

Juglone, a natural toxin derived from the walnut tree, is commonly employed to induce oxidative stress *in C. elegans* (*Kittimongkolsuk et al., 2021*). In this experiment, synchronized worms were treated with CGE at 20, 60, and 100 µg/mL for 3 days. Following treatment, the worms were thoroughly washed three times with M9 buffer and subsequently placed in NGM supplemented with juglone at a concentration of 200 µM to induce oxidative stress. Living nematodes were monitored every 30 min until all specimens had succumbed. Each experimental group comprised about 40 worms, and the entire procedure was replicated three times to ensure reliability. Doses (20, 60, 100 µg/mL) were selected based on logarithmic intervals used in analogous studies of fungal extract bioactivity (*e.g.*, *Tang et al., 2022*), which optimize dose–response analysis within non-toxic ranges and fall within the non-toxic window (10–200 µg/mL) established in our acute toxicity assays.

### Ultraviolet-B stress assay

The worms were subjected to various concentrations of CGE, specifically 20, 60, and 100 µg/mL for three days. After treatment, the nematodes were rinsed three times and subsequently transferred to a sterile NGM plate. Each group, consisting of about 30 worms, was then exposed to ultraviolet light (UV-B) at an intensity of 120 mJ/cm$^2$. Living nematodes were monitored hourly until all individuals had perished. This experimental procedure was repeated three times to ensure robust results.

## Intracellular malondialdehyde content, as well as the activities of superoxide dismutase, catalase, and glutathione peroxidase

Synchronized worms were treated with CGE (20, 60, and 100 µg/mL) for 3 days. The *C. elegans* exposed to three mM $H_2O_2$ solution for oxidative stress for 1 h, and then were cleaned with M9 buffer to get the worms with oxidative stress. The non-stressed worms were not subjected to this step of stress treatment. The worms were disrupted using an ultrasonic cell crushe, and the supernatant was diluted with normal saline, for subsequent measurement. The levels of malondialdehyde (MDA) and protein content, as well as the activities of superoxide dismutase (SOD), catalase (CAT), and glutathione peroxidase (GSH-Px) under oxidative stress and stress-free conditions were determined using commercial assay kits (Nanjing Jiancheng Biotechnology Institute, China). Final results were normalized to protein levels.

## Measurement of reactive oxygen species levels

Synchronized *C. elegans* were subjected to treatment with CGE (20, 60, and 100 µg/mL) for three days. Following this treatment, worms were classified into two groups: those subjected to oxidative stress and those maintained under non-stress conditions, in accordance with the methodology outlined in 'Intracellular malondialdehyde content, as well as the activities of superoxide dismutase, catalase, and glutathione peroxidase'. DCFH-DA (2′,7′-dichlorofluorescin diacetate) was used as a probe to measure intracellular reactive oxygen species (ROS) levels. Worms were incubated with DCFH-DA (diluted to 10 µM with M9 buffer) at 20 °C for 50 min, and then washed with M9 buffer to eliminate excess

DCFH-DA. *C. elegans* were anesthetized with 60 μg/mL levamisole hydrochloride and transferred to 2% agar pads. The study was carried out three times, involving 10 worms per group. The nematodes were examined using a fluorescence microscope (CX23; Olympus, Tokyo, Japan), ensuring consistent exposure times throughout the observations. Relative fluorescence levels were quantified and analyzed by ImageJ.

## Assessment of lipofuscin accumulation and body length

Lipofuscin is one of the important aging markers, which can be self-fluorescent and accumulate gradually with the increase of age, and is a common indicator to evaluate the health status of nematodes (*Kittimongkolsuk et al., 2021*). The synchronized worms were treated with CGE (20, 60, and 100 μg/mL) for 5 days. Worms were anesthetized with 60 μg/mL levamisole hydrochloride and transferred to 2% AGAR pads. The experiment was conducted three times, with each group consisting of ten worms. Exposure times were kept consistent, and observations were made using a fluorescence microscope (CX23, Olympus, Tokyo, Japan). The relative fluorescence and nematode length were analyzed by ImageJ software.

## Movement assay

Referring to the methods of *Tang et al. (2022)* and *Chen et al. (2022)*, the synchronized nematodes were treated with CGE (60 μg/mL) and the motility of nematodes was evaluated at days 3, 7 and 11. Nematodes were transferred to freshly prepared agar plates to facilitate free locomotion for 1 min. Subsequently, the movement behavior of nematodes was observed with a stereomicroscope and divided into three categories: A, B and C. Class A can move spontaneously and smoothly. Class B nematodes move when stimulated, and the tracks they leave behind are non-sinusoidal; Class C nematodes do not move forward or backward, but their noses or tails wiggle when touched. Ten nematodes were assessed in each group, with the experiment being conducted in triplicate.

## RNA-seq

The synchronized nematodes were treated with CGE (60 μg/mL) for 3 days, nematodes with oxidative stress were obtained according to the method described in 'Intracellular malondialdehyde content, as well as the activities of superoxide dismutase, catalase, and glutathione peroxidase', which were set as the experimental group, and the three groups of replicates were named CGE1, CGE2, CGE3, respectively; the control group was the nematodes without CGE treatment, and the three groups of replicates were named CK1, CK2, CK3, respectively. The six groups of samples were promptly frozen using liquid nitrogen for 10 min and then transported with dry ice to Suzhou Panomic Biomedical Technology Co., Ltd. for transcriptome sequencing analysis (using the Illumina sequencing platform.). The statistical power of this experimental design, calculated in RNASeqPower is 0.8. GO (Gene Ontology) and KEGG (Kyoto Encyclopedia of Genes and Genomes) pathway enrichment analyses were performed to characterize the biological functions and signaling pathways of differentially expressed genes (DEGs).

## Data analysis

Survival curves were generated employing log-rank (Mantel-Cox) tests, while other statistical analyses utilized one-way ANOVA followed by *post hoc* comparisons using least significant difference (LSD) and Duncan tests (conducted with SPSS software, version 27.0). All data are presented as mean $\pm$ standard deviation ($n = 3$), with distinct letters in columns signifying statistically significant differences ($P < 0.05$).

# RESULTS

## Analysis of acute toxicity of CGE to *C. elegans*

The acute toxicity of varying concentrations of CGE on nematodes was assessed, and the results are provided in Table S1. In comparison to the control group, the survival rate of nematodes treated with CGE at 10 to 200 μg/mL for 24 h remained unaffected. Specifically, the survival rate of nematodes exposed to the highest tested CGE concentration (200 μg/mL) was 98 $\pm$ 1%. These findings indicate that CGE within this concentration range did not induce acute toxicity in the nematodes. Therefore, CGE in the range of 10–200 μg/mL was selected for further analysis in subsequent experiments.

## Effect of CGE on stress in *C. elegans*

Following treatment of nematodes with various concentrations of CGE for 3 days, the lifespan of the nematodes was examined under diverse stress conditions. The corresponding results are displayed in Fig. 1 and Table S2. Under UV and oxidative stress conditions, the nematode survival curve was shifted to the right by CGE treatment at three concentrations, compared with the control group (Figs. 1A, 1B). The mean and median lifespan of nematodes in the 20, 60 and 100 μg/mL CGE treatment groups were significantly increased under ultraviolet conditions (Table S2, $P < 0.05$), and the mean lifespan was increased by 6.71%, 10.65%, 14.78%, respectively, compared with the control group. The mean and median lifespan of *C. elegans* were significantly increased under juglone-induced oxidative stress (Table S2, $P < 0.05$), and the mean life span was increased by 19.72%, 22.85% and 33.84%, respectively, compared with the control group. These findings suggest that CGE treatment can improve the stress resistance of nematodes. CGE exhibited the optimal protective effect on nematodes under UV and oxidative stress conditions at a concentration of 100 μg/mL.

## Effect of CGE on malondialdehyde content and antioxidant enzyme activity in *C. elegans*

As illustrated in Figs. 2A and 2E, under no stress conditions, the MDA content in *C. elegans* treated with 100 μg/mL CGE decreased by 39% ($P < 0.05$) compared with the blank control group. Under oxidative stress, MDA levels in nematodes treated with 60 μg/mL CGE decreased by 33% ($P < 0.05$). Figures 2B and 2F show that SOD activity increased by 14% ($P < 0.05$) and 10% ($P < 0.05$) in the 20 μg/mL CGE under no stress and oxidative stress conditions, respectively. According to Figs. 2C and 2G, GSH-Px activity increased by 91% ($P < 0.05$) and 17% ($P < 0.05$) with 60 μg/mL CGE under no stress and oxidative stress, respectively. Figures 2D and 2H demonstrate that CAT activity in *C. elegans* increased

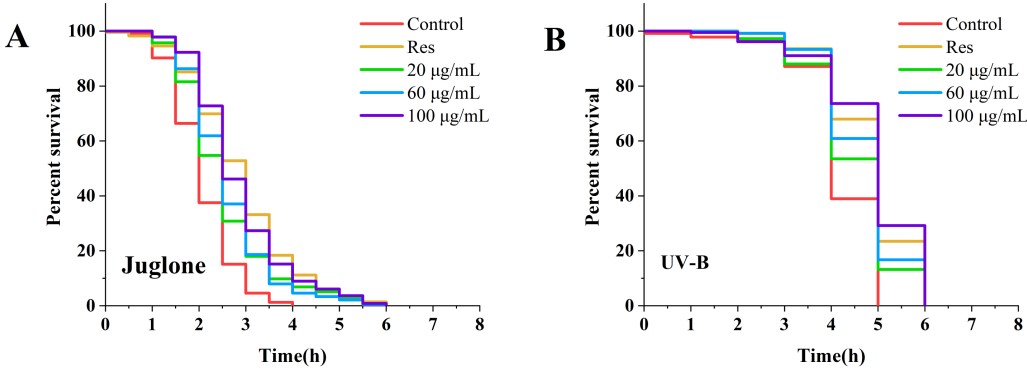

**Figure 1** **The effect of CGE on stress resistance in *C. elegans*.** (A) Survival curve of worms under Juolone-induced stress. (B) Survival curve of worms under UV irradiation-induced stress. "Res" represents 20 µg/mL resveratrol, while 20, 60, and 100 µg/mL specify the concentrations of CGE.

by 110% ($P < 0.05$) after 60 µg/mL CGE treatment and by 48% ($P < 0.05$) following 100 µg/mL CGE treatment under oxidative stress. However, not all concentrations showed significant differences in their effects on the antioxidant enzyme system. These findings suggest that specific concentrations of CGE can activate the antioxidant defense system of *C. elegans*, demonstrating its *in vivo* antioxidant capacity.

## Impact of CGE on the accumulation of reactive oxygen species in *C. elegans*

Levels of reactive oxygen species (ROS) in *C. elegans* were assessed under both non-stress and oxidative stress conditions, with the results depicted in Fig. 3. The data presented in Figs. 3B and 3C illustrate that ROS levels in nematodes exposed to oxidative stress were significantly elevated in compared to those in unstressed nematodes, suggesting that $H_2O_2$ facilitated ROS accumulation in the nematodes. In both cases, ROS levels in the three concentrations of CGE treated nematodes were significantly decreased compared with the blank control group ($P < 0.05$), which was consistent with the fluorescence reduction. In the 20, 60, 100 µg/mL CGE treatment groups, ROS levels were decreased by 25.28%, 30.30%, 22.72% compared with the blank control group under no stress, and 5.43%, 9.94%, 11.13% compared with the blank control group under oxidative stress, respectively. These results indicate that CGE can reduce ROS levels in nematodes under non-stress conditions and $H_2O_2$-induced oxidative stress conditions. 60 µg/mL CGE under non-stress conditions had the strongest ability to reduce ROS levels in nematodes, while CGE of 100 µg/mL under oxidative stress conditions has the strongest ability to reduce ROS levels in nematodes. It is apparent that CGE exhibits a robust antioxidant capacity, effectively reducing intracellular ROS levels. This observation aligns with the findings that CGE treatment enhances the antioxidant stress resilience of nematodes and bolsters their antioxidant defense system.

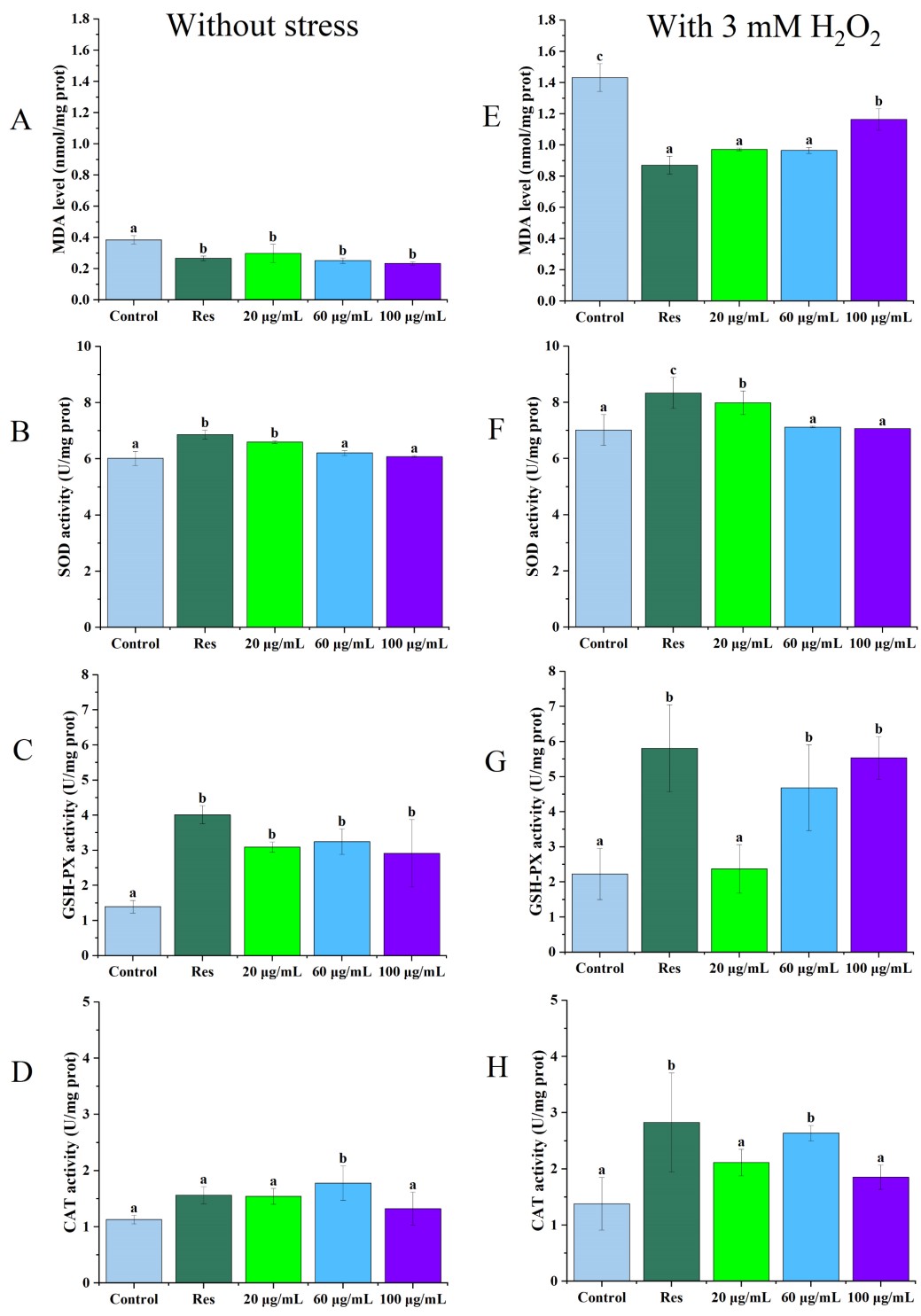

**Figure 2** **The influence of CGE on the antioxidant defense mechanism in *C. elegans* under stress-free conditions and oxidative stress induced by H₂O₂.** (A, E) The MDA level. (B, F) The SOD activity. (C, G) The GSH-PX activity. (D, H) The CAT activity. A D is the result under Stress-free condition, E H is the result under oxidative stress condition. "Res" represents 20 μg/mL resveratrol, while 20, 60, and 100 μg/mL specify the concentrations of CGE. The difference of column shape without common letters was statistically significant ($p < 0.05$).

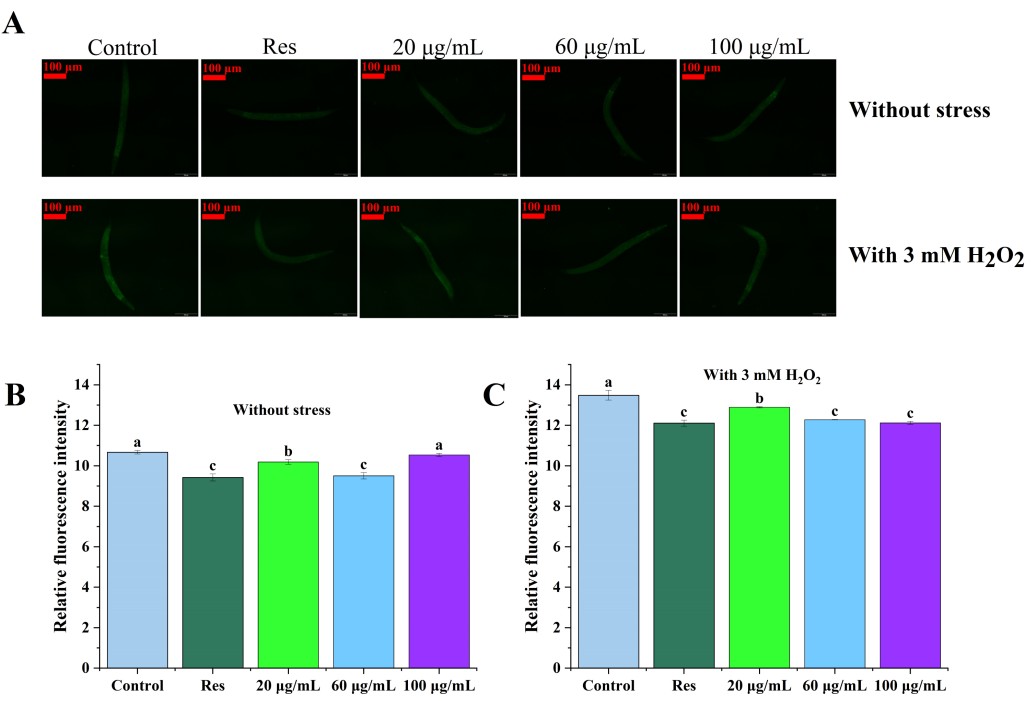

**Figure 3** **The effect of CGE on intracellular levels of ROS in *C. elegans*.** (A) Representative image of worms which were treated with CGE in Stress-free condition and oxidative stress conditions; (B) Accumulation of ROS in *C. elegans* under Stress-free condition. (C) Accumulation of ROS in *C. elegans* under $H_2O_2$-induced oxidative stress. "Res" represents 20 μg/mL resveratrol, while 20, 60, and 100 μg/mL specify the concentrations of CGE. The difference of column shape without common letters was statistically significant ($p < 0.05$).

## Effects of CGE on body length and lipofuscin accumulation in *C. elegans*

The lipofuscin levels of *C. elegans* were further evaluated, as shown in Fig. 4. Compared with the blank control group, 20, 60, and 100 μg/mL CGE treatment increased the body length of nematodes by 6.03%, 11.92%, and 8.12%, respectively (Fig. 4B, $P < 0.05$). Relative fluorescence quantitation showed that compared with the control group, 20, 60, and 100 μg/mL CGE treatment significantly reduced the lipofuscin content in *C. elegans* by 11.23%, 13.99%, and 11.59%, respectively (Fig. 4C, $P < 0.05$). These findings suggest that CGE significantly diminishes lipofuscin accumulation in *C. elegans*, and 60 μg/mL of CGE has the strongest effect on lipofuscin accumulation.

## Effect of CGE on the movement of *C. elegans*

Figure 5 illustrates the influence of CGE on the locomotion abilities of nematodes. As nematodes age, their locomotion capacity decreases, and Class B and C locomotion patterns start to emerge in the intermediate and final phases of the life cycle. However, exercise tests at different life cycle stages did not demonstrate a significant difference in motor ability between the CGE treatment group and the control group, suggesting that CGE did not have a significant impact on the nematodes' motor system.

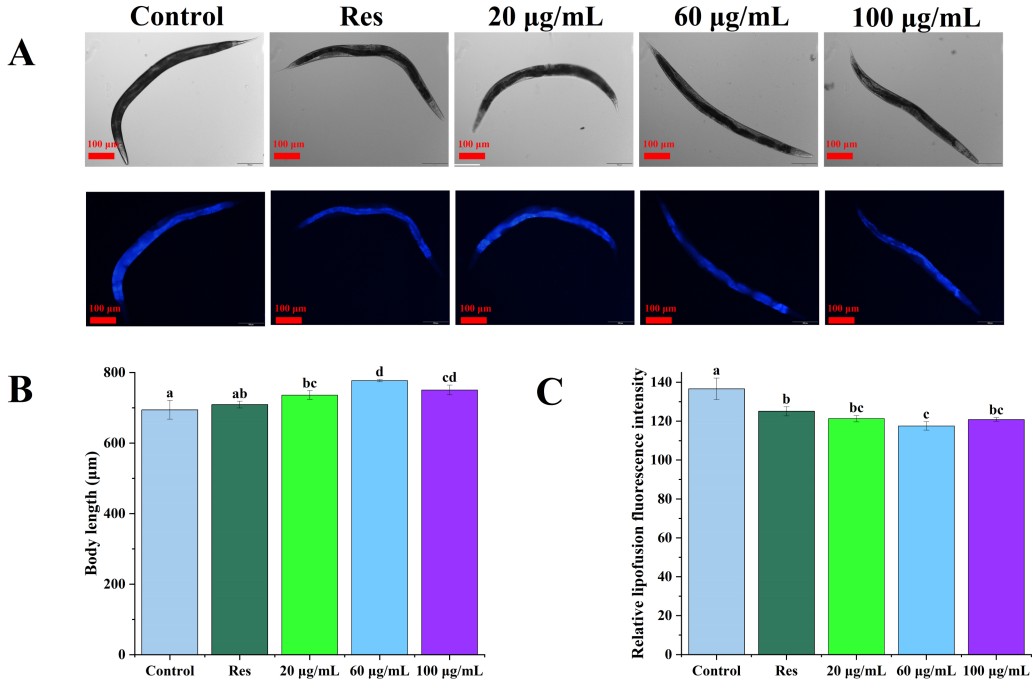

**Figure 4** **The effect of CGE on lipofuscin accumulation and body size in *C. elegans*.** (A) Representative images of fluorescence and bright field micrographs are shown, the scale bar was 100 μm; (B) body length; (C) lipofuscin level. "Res" represents 20 μg/mL resveratrol, while 20, 60, and 100 μg/mL specify the concentrations of CGE. The difference of column shape without common letters was statistically significant (*p* < 0.05).

## Effects of CGE on the transcriptome of *C. elegans* under oxidative stress

### Sample correlation test and differential expression analysis

A coefficient value ranging between 0.8 and 1 indicates a strong correlation, while a value below 0.8 suggests a weak correlation between experimental replicates. Figure 6A shows that the correlation coefficient between the repeat groups of CK group and CGE group is greater than 0.9, indicating that the correlation between the repeat groups is very strong, and the transcriptome data obtained in this experiment has good repeatability and reliability. We used DESeq for differential gene expression analysis, setting the criteria for differentially expressed genes (DGEs) as |log2FoldChange|>1 and *P* < 0.05. Cluster analysis of DGEs across different treated samples (Fig. 6C) revealed significant differences in gene expression between the control and CGE-treated groups, suggesting that CGE treatment alters gene expression patterns in *C. elegans*, potentially enhancing their resistance to oxidative stress.

### GO enrichment analysis of DGEs

The ten GO terms with the lowest *p*-values, reflecting the highest levels of enrichment in each GO category, were selected for presentation. The outcomes are depicted in Fig. 7. In molecular function, DGEs were mainly concentrated in extracellular ligand-gated ion channel activity, polymerase II transcription regulatory region and channel activity. In

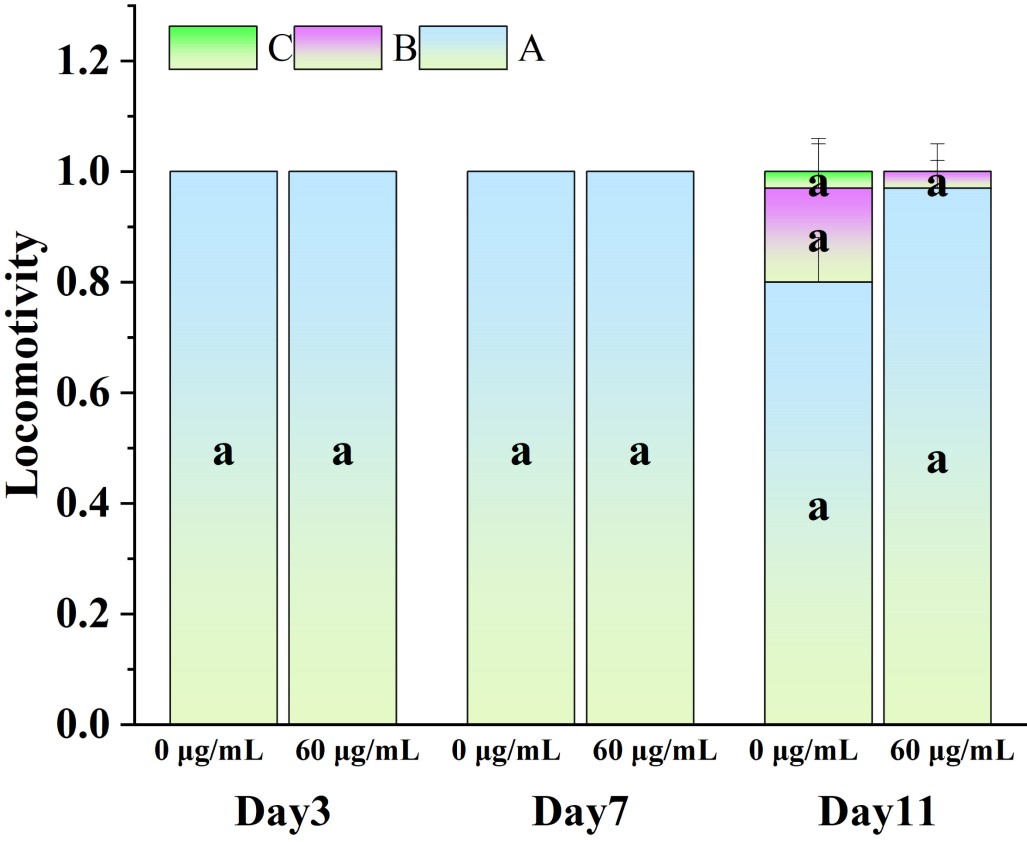

**Figure 5** **The effect of CGE on movement in *C. elegans*.** The concentrations of 0 μg/mL (control) and 60 μg/mL were specified for CGE. The lowercase letter a on the column represents the significance result, and there is no significant difference between the test results among the groups, so the letters are the same ($p > 0.05$).

biological process, DGEs was mainly enriched in the regulation of neurogenesis, regulation of neuron differentiation and cell wall macromolecule catabolic process. In cellular component, DEGs were mainly concentrated in cell junction, synapse and postsynapse. The results indicate that treatment with DGE significantly modulated the expression levels of a range of functional genes in *C. elegans*.

### KEGG enrichment analysis of DGEs

The results of Kyoto Encyclopedia of Genes and Genomes (KEGG) enrichment analysis by DGEs are depicted in Fig. 8, showcasing the top 20 KEGG pathways with the lowest false discovery rate (FDR) values. The analysis highlighted the MAPK signaling pathway within the 'Environmental Information Processing' category as the most significantly enriched pathway in the DGEs. The second most significant enrichment is longevity regulating pathway-multiple species belonging to organismal systems. The third significant enrichment was glutathione metabolism under the category of metabolism. Furthermore, DGEs were significantly enriched in various metabolic pathways, including fatty acid degradation and d-amino acid metabolism. The gene *hsp-70*, which is associated with the

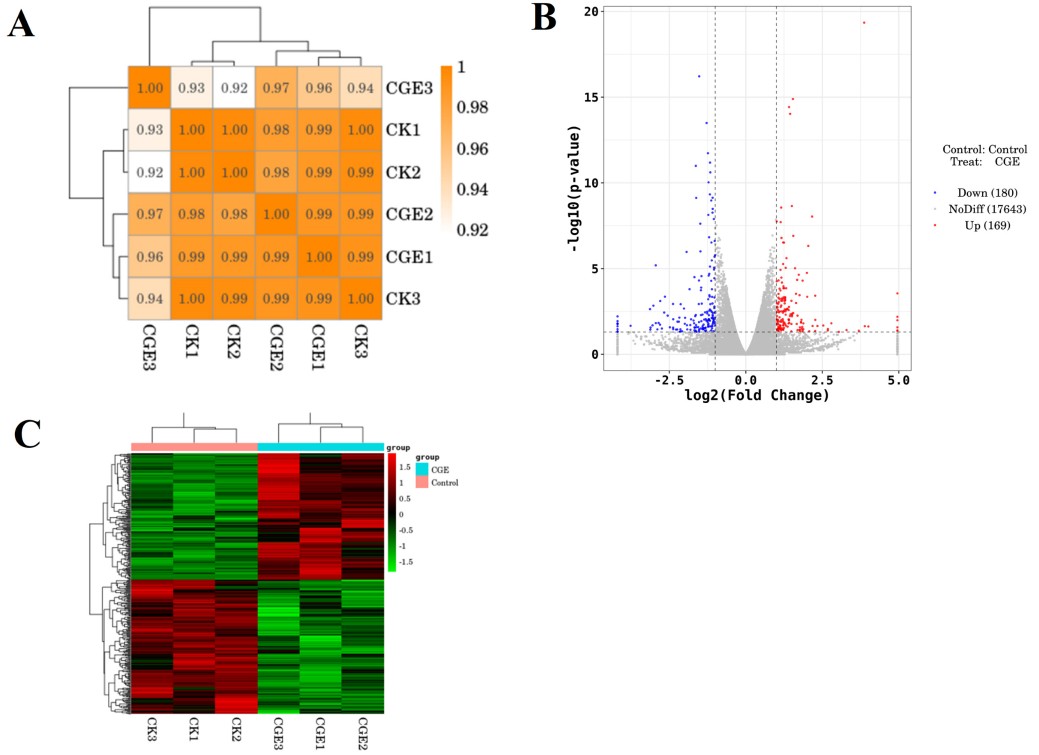

**Figure 6  Sample correlation test and differential expression analysis.** (A) Heat map of expression correlation. The left side and the top side are sample clustering, the right side and the bottom side of the figure are sample names, and squares of different colors represent the correlation level of two samples; (B) volcano map of differentially expressed genes. Red dots indicate up-regulated genes, blue dots indicate down-regulated genes, and gray dots indicate non-significant differentially expressed genes; (C) heat map of clustering of differentially expressed genes. Red indicate high expression genes and green indicate low expression genes.

MAPK signaling pathway and the longevity regulating pathway-multiple species pathways, was markedly up-regulated. The DGEs *gsto-2* (FC = 16.05) and *gst-33* (FC = 14.6) related to glutathione metabolism are the second and fourth genes with the highest upregulation ratio. DGE *acox-3* associated with fatty acid degradation pathway is significantly down-regulated. Significant enrichment of the above metabolic pathways and signaling pathways and expression of related genes, These results suggest that CGE exerts its role in enhancing antioxidant capacity by influencing the metabolism, longevity regulating pathway-multiple species and MAPK signaling pathway.

## DISCUSSION

Numerous studies have demonstrated a direct link between oxidative stress and a range of diseases and aging. The consumption of exogenous antioxidants is crucial in both disease prevention and treatment, as well as in anti-aging strategies (*Liguori et al., 2018*; *Neha et al., 2019*). It is increasingly important to find safe and effective natural antioxidants. Previous studies have demonstrated that CGE exhibits strong antioxidant capacity *in vitro*; however,

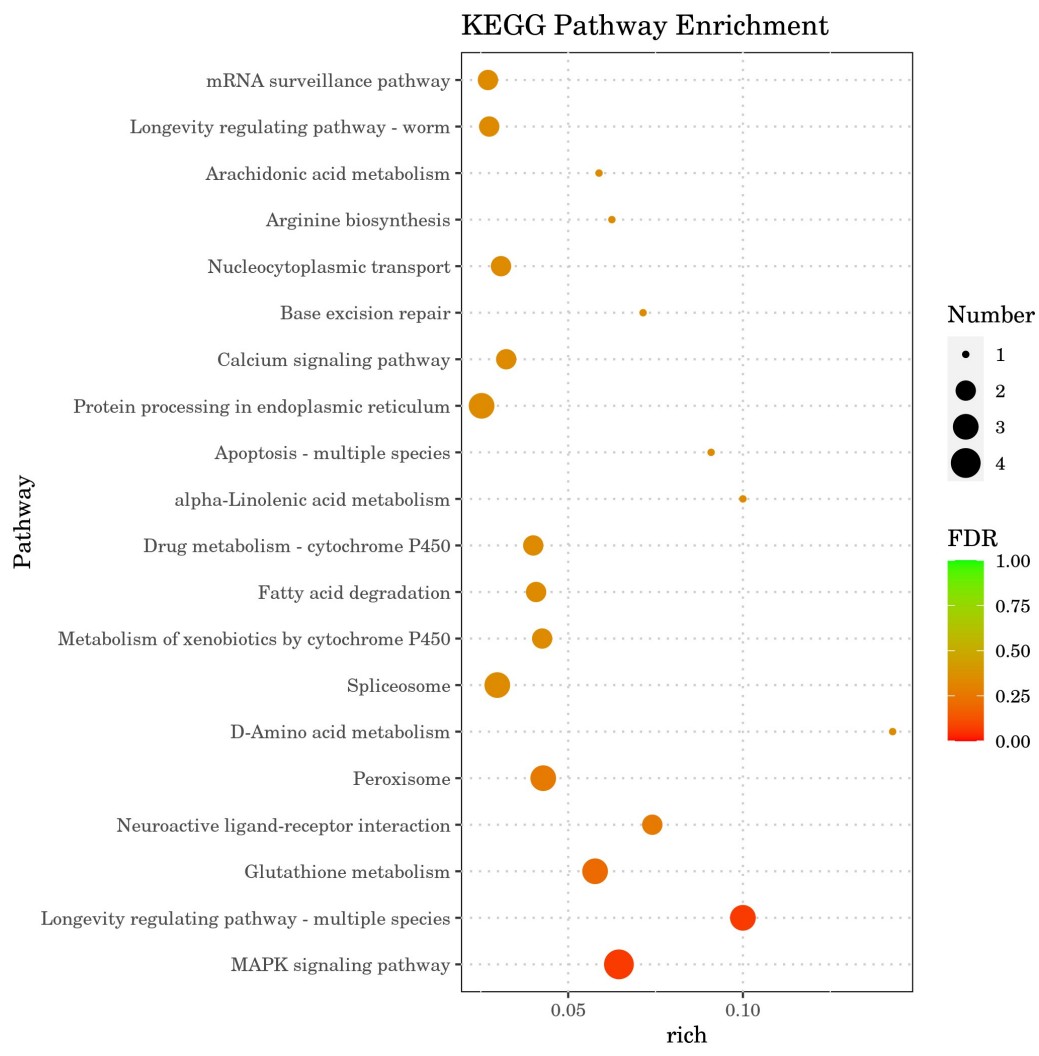

**Figure 7 GO enrichment analysis bar diagram.** The horizontal coordinate is the term of Go level2, and the vertical coordinate is the -log10 (*p*-value) enriched by each term.

its *in vivo* antioxidant activity has not been investigated. To explore this, we employed the model organism *C. elegans*, widely used in antioxidant research. Various stresses in daily life, including oxidative, heat, and ultraviolet stress, can disrupt the balance between oxidation and the antioxidant system, resulting in excessive reactive oxygen species (ROS) production and negatively impacting human health (*Picardo & Dell'Anna, 2010*; *Slimen et al., 2014*). Nematode resistance to environmental stress is generally proportional to lifespan, significantly influencing their longevity, the nematode antioxidant model usually exposed nematodes to different stress environments (juglone, $H_2O_2$, paraquat, ultraviolet, *etc.*) to observe whether the addition of extracts activated the expression of related genes in nematodes and whether it improved the survival rate of nematodes (*Zhu et al., 2022*; *Moreno-Arriola et al., 2014*; *Link, 2006*). This study demonstrated that the experimental concentration of CGE treatment enhances nematode resistance to oxidative and ultraviolet

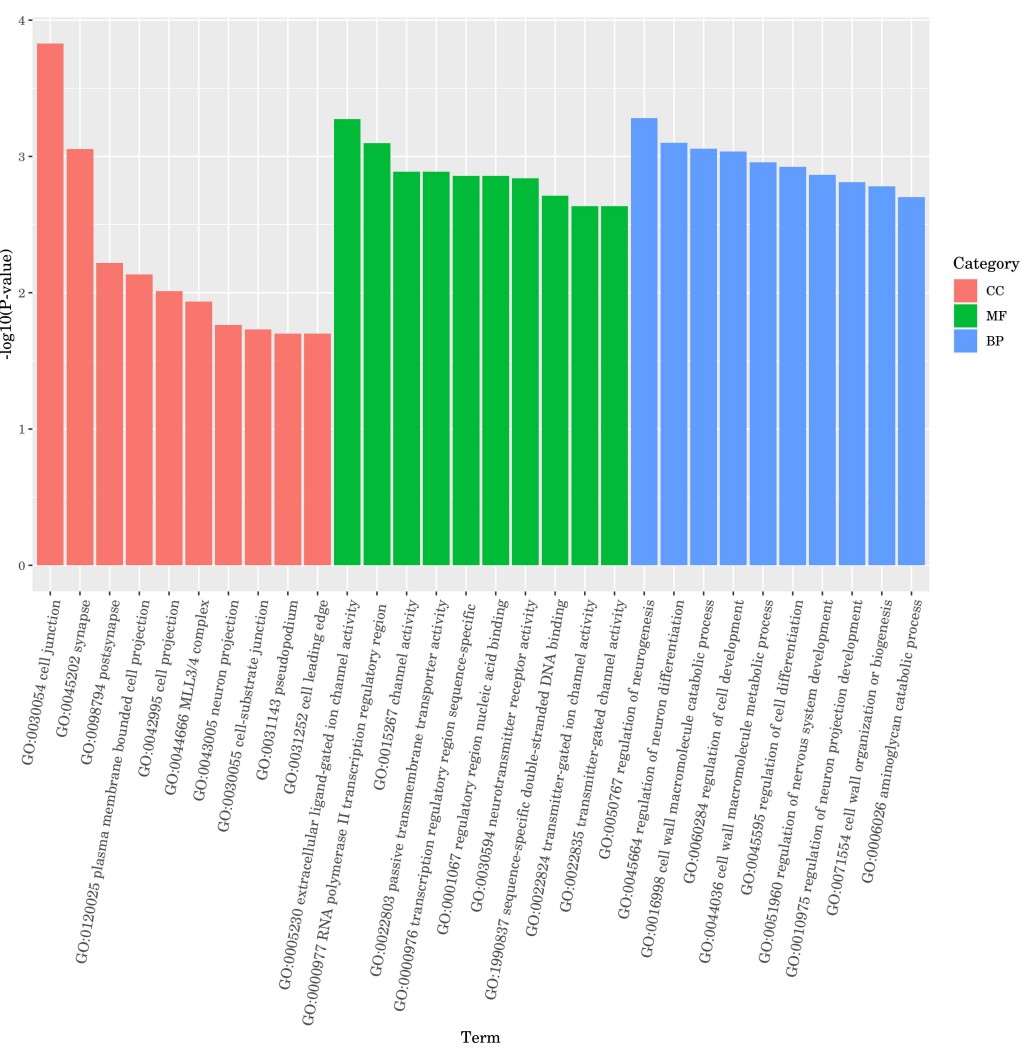

**Figure 8 KEGG pathway enrichment analysis.** The greater the Rich factor, the greater the degree of enrichment. The closer the FDR is to zero, the more significant the enrichment.

stress by activating the antioxidant enzyme system. Specifically, it increases the activities of SOD, CAT, and GSH-Px under both normal and $H_2O_2$-mediated oxidative stress conditions, while reducing ROS levels and MDA content. These results indicate that CGE has good antioxidant activity *in vivo*, aligning with the results of numerous previous studies. Blueberry extract, *Momordica saponin* extract, betulinic acid, *Paecilomyces variotii* extract and resveratrol have been confirmed to improve the anti-stress ability of nematodes, delay aging, and have antioxidant activity *in vivo* (*Chen et al., 2022*; *Wang et al., 2022b*; *Wang et al., 2018*; *Lin et al., 2021*; *Chen, Rezaizadehnajafi & Wink, 2013*).

Locomotion behavior, body length and lipofuscin are commonly used indicators to evaluate the health status of *C. elegans*. Studies have shown that longevity in many species is closely related to body size, locomotion and other indicators, and *C. elegans* shows the same aging performance as other organisms (*Wang et al., 2018*; *Zeng et al., 2019*). As age

progresses, various physiological indicators will show different trends, and its locomotion ability will decrease significantly, and lipofuscin will gradually accumulate. The findings of this study demonstrated that the experimental concentration of CGE treatment was effective in reducing the lipofuscin levels in nematodes; however, it did not influence their locomotion ability. In addition, CGE treatment significantly increased the body length of nematodes, which was consistent with previous studies. *Wang et al. (2022b)* found that the extract of endophytic fungus *Paecilomyces* continued to increase the body length and body width of nematodes. *Zeng et al. (2019)* found that the body length of nematodes increased significantly after Liangyi Gaov was treated, suggesting that the anti-aging effect of Liangyi Gao may also depend on growth. These results indicate that CGE can promote healthy longevity of nematodes.

The effects of CGE on the anti-stress ability, the activity of antioxidant oxidase system and the content of lipofuscin were not concentration-dependent. CGE (100 μg/mL) significantly improved the survival rate of nematodes under UV and oxidative stress, reduced ROS levels during oxidative stress, and increased GSH-Px activity in both stress-free and stressed conditions. CGE with a concentration of 60 μg/mL had the best effect on reducing the ROS content under no stress and oxidative stress, reducing lipofuscin accumulation, and enhancing CAT activity under no stress and oxidative stress. The concentration of 20 μg/mL of CGE had the best effect on enhancing SOD activity of nematodes under non-stress and oxidative stress conditions. Overall, the CGE effect of 60 μg/mL is slightly higher than that of 100 μg/mL. This result may be caused by the complex antioxidant enzyme system combined with the complex chemical composition of CGE. Some compounds in CGE, such as its main component caffeic acid, have stimulating reactions, and when the concentration is increased to a certain level, the effect on nematodes will be weakened or even harmful (*Pietsch et al., 2011*). The findings of *Li et al. (2021)* demonstrated that 300 μM of caffeic acid (54.048 μg/mL) enhances nematode resistance to heat stress and paraquat-induced oxidative stress, which aligns with the results of this study. This indicates that caffeic acid may be a key contributor to the antioxidant capacity of CGE.

In order to further study the molecular mechanism of CGE affecting nematodes, transcriptome sequencing of nematodes was conducted. Gene expression difference analysis and enrichment analysis showed that CGE treatment affected the functional gene expression of nematodes, and significantly affected the genes related to the MAPK signaling pathway, longevity regulation pathway, glutathione metabolism and Fatty acid degradation pathway of nematodes, which may enhance the antioxidant capacity of nematodes by regulating these pathways. MAPK signaling pathway is a key component of *C. elegans* immune response, which plays a key role in cell proliferation, differentiation, transformation and apoptosis, and is closely related to inflammation, tumor and other diseases (*Xu et al., 2023*; *Kim, Sohn & S.Lee, 2022*). *Sugawara, Saraprug & Sakamoto (2019)* found that soy sauce may enhance the oxidative stress tolerance of nematodes through the p38 MAPK pathway (an important group of signaling molecules in the MAPK family). The DGEs in the CGE group were most significantly enriched in the MAPK signaling pathway, indicating that CGE has a significant regulatory effect on the MAPK signaling pathway of nematodes and exerts its effects on nematodes through this pathway. The DGE *hsp-70* is a

member of the heat shock transcription factor family, which can regulate the heat shock response. The heat shock protein is involved in the regulation of aging and longevity in a variety of organisms, and can protect cells from the influence of harmful stimuli, playing a prominent role in heat stress and oxidative stress (*Hsu, Murphy & Kenyon, 2003*; *Lu et al., 2020*). The research results of *Lu et al. (2020)* showed that phytoestrogens SDG may be mediated by HSF-1, which can regulate the expression of small heat shock protein HSP-16, increase the stress resistance of nematodes and delay aging. *Zhang et al. (2024)* found that *Polygonatum cyrtonema* polysaccharide can up-regulate the expression level of hsp-16.2 gene in *C. elegans*, and can enhance the oxidation defense and stress resistance of *C. elegans*. Therefore, it is speculated that CGE up-regulation of *hsp-70* expression is one of the reasons for enhancing the resistance of nematodes to oxidative stress. Significantly upregulated *gsto-2* and *gst-33* encode glutathione-S-transferase, and the glutathione transferase family has an important detoxicating effect on ROS stimulation, can catalyze the binding of exogenous and endogenous compounds to GSH, and is mainly involved in inhibiting oxidative stress (*Moreno-Arriola et al., 2014*; *Wang et al., 2022a*; *Zhang et al., 2022a*) any studies have shown that GST-4 of this family plays an important role in scavenging free radicals and enhancing oxidative stress. *Urbizo-Reyes et al. (2022)* showed that canary seed peptide fraction (CSPF) relies on upregating GST-4 of *C. elegans* to enhance its ability to remove free radicals. Consequently, the findings of this study suggest that the enhanced antioxidant capacity of *C. elegans* treated with CGE may be closely associated with the notable up-regulation of *gsto-2* and *gst-33*. The DGE *acox-3* encodes the major acyl-CoA oxidase ACOX-3, which is involved in fatty acid regulation (*Qin, Wang & Chu, 2021*; *Gao et al., 2021*). *Sun et al. (2023)* found that ginsenoside residues (GRP) can up-regulate the expression level of *acox-5*, promote the degradation of fatty acids in *C. elegans*, and induce the preferential synthesis of beneficial fatty acids, GRP may delay the aging of *C. elegans* by affecting the biosynthesis of fatty acids. It was speculated that CGE could improve the health status of nematodes by regulating the pathway of fatty acid degradation, thus enhancing the stress resistance of nematodes.

In order to more accurately analyze the molecular mechanisms that influence CGE on nematodes, qRT-PCR, WB, nuclear localization and other methods can be used to validate the key functions selected by transcriptome analysis. In addition, the application of CGE in the food and medical fields also needs to be validated with more advanced animal models, and more in-depth studies of *in vivo* activity and safety.

## CONCLUSION

CGE improves nematode stress resistance, enhances antioxidant enzyme activity under both basal and stress conditions, reduces MDA, ROS, and lipofuscin accumulation, increases body length, and does not affect nematode locomotion ability. Under oxidative stress, CGE plays a role in enhancing antioxidant capacity by regulating genes related to the glutathione metabolism pathway, the fatty acid degradation pathway, the MAPK signaling pathway and longevity regulation pathway. CGE has the potential to be developed as an antioxidant

in the food and medical fields, and the endophytic fungus *C. globosum* has good potential to be developed as a natural antioxidant source.

### Funding
The authors received no funding for this work.

### Competing Interests
The authors declare there are no competing interests.

### Author Contributions
- Nayu Shen conceived and designed the experiments, performed the experiments, prepared figures and/or tables, and approved the final draft.
- Zhao Chen performed the experiments, prepared figures and/or tables, and approved the final draft.
- Siyu Wang performed the experiments, prepared figures and/or tables, and approved the final draft.
- Mingqi Zhang performed the experiments, prepared figures and/or tables, and approved the final draft.
- Yujie Jia analyzed the data, prepared figures and/or tables, and approved the final draft.
- Xinyu Zhang analyzed the data, prepared figures and/or tables, and approved the final draft.
- Yirong Xiao analyzed the data, prepared figures and/or tables, and approved the final draft.
- Zizhong Tang conceived and designed the experiments, prepared figures and/or tables, authored or reviewed drafts of the article, and approved the final draft.
- Qingfeng Li analyzed the data, authored or reviewed drafts of the article, and approved the final draft.
- Ming Yuan analyzed the data, authored or reviewed drafts of the article, and approved the final draft.
- Tongliang Bu analyzed the data, authored or reviewed drafts of the article, and approved the final draft.

### Data Availability
The raw measurements are available in the Supplementary Files.

### Supplemental Information
Supplemental information for this article can be found online at http://dx.doi.org/10.7717/peerj.19827#supplemental-information.

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
