# Peer review of "Chaetomium globosum from Alisma orientale (Sam.) Juzep. enhances the antioxidative stress capacity of Caenorhabditis elegans"

_PeerJ, doi:10.7717/peerj.19827_

## Round 0.1 · original submission · Minor Revisions

Please address the reviewers' suggestions and provide a revised version.

**Language Note:** The review process has identified that the English language must be improved. PeerJ can provide language editing services - please contact us at [email protected] for pricing (be sure to provide your manuscript number and title). Alternatively, you should make your own arrangements to improve the language quality and provide details in your response letter. – PeerJ Staff

Reviewer 1 ·

Basic reporting

The manuscript reports interesting data and is properly written.

Experimental design

The experiments are sound, but some improvements can be made to the methodology (see section 4 for specific comments).

Validity of the findings

The data and conclusions are sound.

Additional comments

Manuscript titled “Chaetomium globosum from Alisma orientale (Sam.) Juzep. enhances the antioxidative stress capacity of Caenorhabditis elegans” reports various analysis about the effects of Chaetomium globosum ethyl acetate extract, on various parameters of the nematode C. elegans. There are some comments and suggestions for the authors:

1. Section 2.1 provides minimal explanation about how the samples were collected. For example, line 86 states that the fungus was isolated from Alisma orientale; please consider providing additional information, such as any selection criteria or any data relevant to sample collection.

2. Line 115 states that CGE was administered at 20, 60 and 100 ug/mL for 3 days. Can the authors provide a brief justification for why they chose these specific treatment conditions? Particularly since the non-toxic range was 10-200 ug/mL, as mentioned in line 190.

3. Please define “DCFH-DA” in line 143.

4. In section 2.9, please provide additional information about the method, equipment, reagents or experimental conditions that could be relevant to mention, since the section only states that the analysis was made but no other information is provided. Some methodological data is specified in section 3.7.1, which may be more appropriate to mention here instead.

5. Sections 3.7.2 and 3.7.3 report results for GO and KEGG enrichment analyses, however, these are not cited in materials and methods, please mention them there.

6. Lines 394-397 of the conclusion repeat “of nematodes” three times, please state this only once.

7. Please remove the declaration in lines 472-473, which appears to be for another journal.

8. In figure 1, both its legend and the figure itself, please change to “Juglone” (a “g” is missing).

9. On the Y axes of figure 2, please change to “mg prot” (add a space between the two words).

10. On the legends of figures 1-5, please define “Res” and indicate what compound the concentrations 20, 60 and 100 ug/mL are referring to. This will make them unambitious and understandable on their own.

·

Basic reporting

In this study, the authors utilized the nematode Caenorhabditis elegans as a model organism to assess the antioxidant capacity of an extract derived from the supernatant of a Chaetomium globosum culture, isolated from the plant Alisma orientale. The extract was prepared using ethyl acetate extraction.
The study was well-conducted and shows promising results regarding the antioxidant capacity of the extract.

Experimental design

The materials and methods were described with sufficient detail and information to allow replication.

Validity of the findings

All the data supporting the study have been provided; they are robust and statistically well analyzed. Likewise, the conclusions are articulated, appropriately connected to the original research question, and limited to the interpretation of the results.

Additional comments

Minor comments:
- Line 60-63: Rewrite and delete repeated sentences.
- Line 91: Indicate the concentration of the extract with respect to the original volume of the culture.
- Lines 130, 140, and 153: Indicate the concentration of CGE used in the experiment
- Lines 263-266: Delete repeated sentences
- Line 275: change “egulation” to “regulation”
- Line 294: Consider changing the word "stress" by "capacity" or "response"

---

## Round 0.2 · accepted · Accept

The authors have addressed all the reviewers' comments.